# Attitudes toward Patient-Centred Care, Empathy, and Assertiveness among Students in Rehabilitation Areas: A Longitudinal Study

**DOI:** 10.3390/healthcare11202803

**Published:** 2023-10-23

**Authors:** Ana Monteiro Grilo, Graça Vinagre, Margarida Custódio dos Santos, Joana Ferreira Martinho, Ana Isabel Gomes

**Affiliations:** 1H&TRC—Health & Technology Research Center, Escola Superior de Tecnologia da Saúde, Av. D. João II, Lote 4.69.01, 1990-096 Lisboa, Portugal; 2CICPSI, Faculdade de Psicologia, Universidade de Lisboa, Alameda da Universidade, 1649-013 Lisboa, Portugal; margarida.santos@estesl.ipl.pt; 3Escola Superior de Enfermagem de Lisboa, Av. Prof. Egas Moniz, 1990-096 Lisboa, Portugal; gvinagre@esel.pt; 4Escola Superior de Tecnologia da Saúde, Av. D. João II, Lote 4.69.01, 1990-096 Lisboa, Portugal; 5Câmara Municipal de Oliveira do Bairro, Ed. Paços do Concelho, Praça do Município, 3770-851 Oliveira do Bairro, Portugal; joanaferreiramartinho@gmail.com; 6Faculdade de Psicologia, Universidade de Lisboa, Alameda da Universidade, 1649-013 Lisboa, Portugal; ana.fernandes.gomes@psicologia.ulisboa.pt

**Keywords:** patient-centeredness, empathy, assertiveness, rehabilitation, person- and technique-oriented professions, undergraduate health students, perception of communication skills

## Abstract

This study assessed attitudes toward patient-centred care, empathy, assertiveness, and subjective perception of communication skills and technical knowledge among Portuguese undergraduate students in healthcare. These students may develop rehabilitation activities with patients in their person-oriented or technique-oriented professions. Portuguese nursing and allied health students from two public higher education schools completed questionnaires in the first and third academic years: Patient-Practitioner Orientation Scale, Jefferson Scale of Physician Empathy, Scale for Interpersonal Behaviour, and a subjective perception of technical knowledge and communication skills. A total of 183 students completed the surveys. In the first year, students showed moderate to high scores on patient-centredness attitudes, empathy, and assertiveness and perceived themselves as having good communication skills. Students from person-oriented programmes significantly improved their Total and shared patient-centred attitudes in the third year compared with students attending technique-oriented professions. Significant differences in empathy were found between groups in the third year. Distress associated with assertive behaviours increased significantly across time in students from technique-oriented programmes compared with their peers in person-oriented programmes. The results suggest that the health profession’s orientation and the programmes’ specific curriculum might have a role in how some dimensions evolved in the two groups of students. The increasing assertiveness-related discomfort highlighted the importance of assessing and monitoring students’ emotional wellbeing during their initial interactions with patients.

## 1. Introduction

Patient-centredness has been described as a core philosophy of healthcare professionals to endorse high-quality healthcare [1,2], and a growing number of studies have shown that person-centred care practices are significantly associated with positive effects on a range of measures, such as higher satisfaction with care, a better quality of life, better rehabilitation outcomes, and lower care costs [3,4]. In a recent scoping study, Jesus et al. [5] proposed a model for person-centred rehabilitation (PCR Model), defining *patient-centred care* (PCC) in rehabilitation as “*a way of thinking about and providing rehabilitation services ‘with’ the person*” (p. 5), and keeping the focus on how care is organized and provided by health professionals and how patients experience it. Empathy is recognized as an essential component of PCC, allowing for an adequate understanding of patients’ feelings and perspectives and facilitating the relationship between health professionals and patients [6,7].

Also, the awareness that being person-centred, having empathy, and being assertive are fundamental requisites to shape effective health communication, and the recognition that these skills could be learned and/or developed, has highlighted the importance of assessing these competencies in undergraduate health students through their academic training. Bejarano et al. [8] conducted a meta-analysis on health students’ (e.g., medicine, nursing, dentistry, speech therapy, chiropractic therapy, and physical therapy) patient-centred attitudes in this field. The authors noticed low attitudes towards patient-centred care, with females showing significantly higher levels than males. Most longitudinal evaluations of patient-centredness attitudes during the academic programme were performed with medical students; the greatest findings pointed out an overall decline in patient-centredness scores as training progressed, especially at the end of clinical training [9,10,11,12,13]. On the contrary, Ross and Haidet [14] observed a significant improvement in physical therapy students’ attitudes toward patient-centred care after their education experience.

Regarding empathy, it includes competence in listening to others respectfully, promoting their speech, and helping to clarify their worries and claims in a caring manner [15]. In a systematic review that included 30 studies with medical students, Andersen et al. [16] found lower levels of empathy in 14 studies, especially in the advancing academic years. In most studies (18), females also presented higher levels of empathy than males. These results were corroborated by two other systematic reviews with medical students [17,18] and residents [18]. In Nunes et al.’s [19] work, a decline in self-reported empathy during the first year of training was also observed in five groups of health programme students (namely, dentistry, pharmacy, medicine, veterinary medicine, and nursing), with medical, nursing, and dental students achieving statistical significance. Recently, Jia-Ru et al. [20] conducted a systematic review and meta-analysis, including 19 cross-sectional studies with 5047 nursing students; the authors found that nursing students worldwide have, overall, higher levels of empathy, especially females.

Finally, assertive communication has been identified as essential to enhance the performance of health teams [21,22,23], enhance patient safety [22,23,24], and reduce anxiety in health professionals [25,26,27]. However, in a sample of 426 Turkish nursing students from 1 to 4 years, Yilmaz et al. [28] found that approximately half (49.3%) were unassertive, and the other half (50.7%) were assertive at a low level. Shrestha [29] observed moderate assertiveness regarding female nursing students in Nepal. Deltsidou’s [30] and Begley and Glacken’s [31] studies showed that nursing students from Greece and Ireland increased their assertiveness in more advanced semesters. More recently, Ben Cherifa et al. [32] noticed that 36.8% of first-year Tunisian medical students were assertive, with females having lower assertive behaviours than males.

### 1.1. Health Professions and Rehabilitation: Orientation towards Person vs. Technique

Although quality criteria for all health professionals involved in rehabilitation care require graduate training in both technical and communication skills, the core work and focus of each healthcare professional may differ according to their involvement with, and responsibilities towards, the patient and their families and their role in the multidisciplinary team [33,34]. For instance, rehabilitation physiotherapists and nurses establish solid and long-lasting relationships with their patients, often with close physical contact. On the other hand, medical imaging technicians or pharmacists contribute to a patient’s rehabilitation by performing diagnostic examinations or providing medical prescriptions, respectively. The earlier literature has sought to study and relate individuals’ traits and characteristics to the type of health profession or medical specialty chosen, being more *person-oriented*, i.e., professions that are more oriented to the patient/person and their families, in which the clinical assessment and treatment depends on the development of a therapeutic relationship, or more *technique-oriented*, i.e., professions that are mainly focused on the development of technical skills or lean towards procedures and technologies for the patient’s assessment and treatment [35,36]. A recent cross-sectional study developed by Blanco Canseco et al. [37] with medical students from the first, third, and sixth academic years found higher levels of empathy in individuals attending general specialties (e.g., internal medicine, psychiatry, paediatrics, or family medicine) compared with those from surgical-technological (e.g., surgery, radiodiagnosis) or non-clinical specialties (e.g., clinical analysis, pathological anatomy, or preventive medicine).

Most studies assessing students’ communication skills are cross-sectional and focused on medicine and nursing professions. Only a minority included other health students [8,38] and assessed these skills throughout the academic progression [16,38]. On the other hand, some studies simultaneously assess students’ empathy and person-centeredness [39]; however, research that integrates Interpersonal behaviour is minimal. Additionally, to our knowledge, the person or technique orientation taxonomy has yet to be adopted when studying dimensions of communication in higher education students who attend nursing and allied health programmes.

### 1.2. Aims of This Study

The present study aims to fill these gaps by assessing attitudes toward patient-centred care, empathy, Interpersonal behaviours, and the perception of communication skills and technical knowledge among Portuguese students from healthcare areas who may develop rehabilitation activities with patients, considering both person- and technique-oriented professions at the beginning of the first year and the end of the third academic year.

## 2. Materials and Methods

### 2.1. Study Design

This study employed a longitudinal design with a quantitative approach. Students were evaluated at two different moments in their academic paths: the beginning of the first and the end of the third bachelor’s year. These time-point measurements were chosen to control students’ exposure to specific contents of the programme curricula, contact with patients and their families in clinical contexts during internships, and clinical training skills across time. At the end of the third year, students from all programmes attended curricular units that included basic communication skill training and already had the opportunity to perform some clinical practice in academic internships.

### 2.2. Participants

Participants were Portuguese nursing and allied health students (i.e., clinical physiology, dietetics and nutrition, medical imaging and radiotherapy, physiotherapy, and pharmacy) from Lisbon public higher education schools. These programmes were chosen considering their relation to health professions involved in patients’ recovery and rehabilitation in diverse specialties.

For data analyses, students were grouped according to the type of programme they were attending, following Campbell and colleagues’ [36] proposed classification based on the core professional activity (*person-oriented* or *technique-oriented*) of allied health professions. As such, clinical physiology, medical imaging and radiotherapy, and pharmacy were categorized as *technique-oriented* professional programmes, while dietetics and nutrition and physiotherapy were classified as *person-oriented* professional programmes. Nursing was also considered a *person-oriented* profession, considering that the patient-centred care approach is widely adopted in nursing healthcare [40,41].

### 2.3. Procedure

The recruitment occurred at the beginning of the 2016/17 academic year (i.e., September 2016); all first-year nursing and allied health students at the two public higher education schools were invited to participate. The study was presented in the first semester’s lesson (study’s objectives, the guarantee of data confidentiality, and voluntary participation), and informed consent was delivered to each student. Those who agreed to collaborate received the questionnaires to be filled out during that lesson; each protocol was sealed in an envelope with the student’s identification number. The filling time was about forty minutes. Students were again contacted at the end of the third academic year (i.e., May/June 2019) to complete a similar evaluation protocol, and the data collection followed the same procedure. The questionnaires were paired at the end of the study, considering the student’s number. This study was approved by the Ethics Committee of the Nursing School of Lisbon (ESEL) (No. 4283/2016).

### 2.4. Instruments

**Sociodemographic questionnaire.** This questionnaire included questions about students’ age, gender, programme, academic year, civil status, and information about the participant’s current situation regarding residence (i.e., if the student was displaced from his/her hometown) and professional status (i.e., working student).

**The Jefferson Scale of Physician Empathy—Student Version (JSPE-S).** The JSPE-S is a self-report inventory with 20 items scored on a 7-point Likert scale (ranging from 1 = *strongly disagree* to 7 = *strongly agree*) that measures health students’ empathy in clinical contexts during patient interaction. Earlier studies showed good internal consistency of the scales (0.89 and 0.87) [42]. Factor analysis also supported the underlying components of the JSPE-S for pharmacy and nursing students [43]. Although the Portuguese version [44] of the scale originally indicated adequate values of internal consistency (0.84), we removed some items (i.e., items 3, 6, 19, 19) to achieve similar psychometric indicators (Cronbach’s alpha of 0.80). Participants’ answers to the items were summed to obtain the total score of the scale; higher scores (ranging from 16 to 112) represent greater empathic orientation.

**Scale for Interpersonal Behaviour (SIB-S).** A short-form version of the Scale for Interpersonal Behaviour (SIB-S) [45] was used to evaluate the discomfort felt when students are required to act assertively (i.e., *Distress*) and the frequency with which one acts assertively in different interpersonal situations (i.e., *Performance*). Like the SIB-S, the 25 items were rated on a 5-point Likert scale, on two separate response scales: one to assess the intensity of discomfort in interaction situations (1 = *nothing* to 5 = *extremely*) and the other to evaluate the frequency of practicing assertive behaviour in these exact situations (from *never* to *always*). The scores were calculated through the sum of the item’s responses in each scale; higher scores indicate higher discomfort (*Distress* scale) or higher frequency (*Performance* scale) when adopting assertive behaviours. The Portuguese version adopted in this study [46] revealed good values of Cronbach’s Alpha in both scales (0.90 for the *Distress* scale and 0.85 for the *Performance* scale); we found similar reliability indicators in our sample (0.92 for the *Distress* scale and 0.85 for the *Performance* scale).

**Patient-Practitioner Orientation Scale (PPOS).** The Patient Practitioner Orientation Scale [47] is one of the most used questionnaires to measure patient-centeredness. The scale can be administered to doctors or patients and includes 18 items divided into two subscales: *Sharing*, i.e., the extent of the respondent’s belief about the importance of sharing information and power as well as the willingness to share control in decision-making, and *Caring*, i.e., the extent of the respondent’s belief about the importance of emotions, good interpersonal relationships, and treating the patient as a whole. The PPOS has been frequently employed in comparable studies across cultures and has revealed good psychometric properties [10,47,48,49,50,51,52,53,54,55,56,57,58]. In our sample, some items were not retained for total score calculation purposes (e.g., items 9, 10, 13, 17) after performing factorial and reliability analyses [59]; the Total scale (14 items) and subscales *Caring* (9 items) and *Sharing* (5 items) showed acceptable internal consistency (Cronbach’s alpha from 0.52 to 0.67). For each item, participants answered on a 6-point Likert scale (1 = *strongly agree* to 6 = *strongly disagree)*. The mean value of the items’ responses was calculated to achieve Total scale and subscale scores; higher scores indicated greater patient-centeredness attitudes.

**Perception of technical knowledge and communication skills.** To evaluate students’ perception of their communication skills and technical knowledge as a future healthcare professional, we used the Portuguese version of an instrument [60] created by Cleland et al. [61], with two statements: “*As a future healthcare professional, I consider that I have good communication skills*” and “*As a future healthcare professional, I consider that I have good technical knowledge*”. Responses were given on an agreement scale from 1 = *strongly disagree* to 5 = *strongly agree*.

### 2.5. Statistical Analysis

The software package IBM SPSS Statistics for Windows, version 27.0, was used to perform the statistical analysis. For analysis purposes, we only retained data from participants who completed the entire evaluation protocol. Missing data were addressed through the median substitution method. We began by running descriptive analyses of all the variables studied. We previously checked all mandatory assumptions for each inferential test; due to unbalanced sample sizes of person- and technique-oriented profession groups, the robustness of the equal variance assumption was used to validate the parametric test analyses. Qui-square and independent *t*-tests (or the Mann–Whitney U test as a non-parametric alternative) were run to compare differences between groups at baseline. We also conducted mixed ANOVA (within–between subjects) to assess the mean differences between person- and technique-oriented groups at the two time point measurements (first and third academic year); Mauchly’s test of sphericity was waived due to the 2 × 2 study design. When the assumptions for mixed ANOVA were not met, the differences in each group across time or between groups at the third academic year were assessed through independent or paired *t*-test (or their non-parametric alternatives, the Mann–Whitney U test or the Wilcoxon Signed-Ranked test).

## 3. Results

In total, 330 nursing and 230 allied health students were invited to participate in the study; 206 students (36.8% response rate) filled out both evaluation protocols, and 183 were considered eligible for analysis (i.e., individuals who completed all questionnaires, each of them for at least 95% of the questions).

### 3.1. Participants’ Characteristics and Person-Oriented vs. Technique-Oriented Groups’ Equivalence at Baseline

Table 1 presents the results regarding the sociodemographic characteristics of the sample at the time of the first survey. Most students were female, single, were not working students, and were not displaced from their usual residence to study. The mean age of the participants was 18.95 (SD = 2.58). We received complete evaluation protocols from students from the six health-related programmes, most of them attending the Nursing programme. No significant differences between the person-oriented and technique-oriented groups were found at baseline regarding the *Displacement from their hometown* variable (χ^2^_(1)_ = 0.044; *p* = 0.834); for the *Sex*, *Marital Status*, and *Working student* variables, the minimum expected count for chi-square tests was not achieved.

Table 2 reports the scores for the *patient-centeredness attitudes, Interpersonal behaviour*, *Empathy*, and *Perceived communication skills and technical knowledge* variables in the first and the third academic years for the whole sample, and discriminately for the person- and technique-oriented groups, respectively. Overall, the whole sample mean scores regarding patient-centredness attitudes in the first academic year were moderate considering the maximum score achievable (six points). Comparatively, the *Caring* subscale mean scores (i.e., importance of treating the patient as a whole beyond their medical condition, valuing patients’ emotions, and good interpersonal relationships) were higher than the *Sharing* subscale mean scores (i.e., importance and willingness to share information, power and control regarding decision-making with patients). Moreover, students had relatively high values in empathy considering the maximum score achievable (112 points). Regarding Interpersonal behaviour, the students’ mean degree of discomfort when performing specific assertive behaviours (i.e., subscale *Distress*) was lower than the mean frequency of practicing the same assertive behaviours in social situations (i.e., subscale *Performance*). Students perceived themselves as having relatively high communication skills and moderate technical knowledge at the beginning of their programme.

When comparing the baseline mean values between the person-oriented and technique-oriented groups regarding all variables assessed, we found that, at the beginning of the first academic year, students from person-oriented programmes had significantly overall higher scores in patient-centeredness (t_(181)_ = 3.345, *p* = 0.001, Cohen’s d = 0.629), Caring (t_(181)_ = 2.955, *p* = 0.004, Cohen’s d = 0.488), and Sharing (t_(181)_ = 2.599, *p* = 0.010, Cohen’s d = 0.555) attitudes than technique-oriented programmes’ students. The students’ groups did not significantly differ in empathy (t_(181)_ = 0.688, *p* = 0.492, Cohen’s d = 0.129) and Interpersonal behaviours (*Distress*: t_(181)_ = 1.637, *p* = 0.103, Cohen’s d = 0.308; *Performance*: t_(181)_ = 0.357, *p* = 0.721, Cohen’s d = 0.067) at baseline. Likewise, students’ perception of their technical knowledge (U = 2190, Z = −1.675, *p* = 0.094) and communication skills (U = 2441, Z = −0.560, *p* = 0.576) at the beginning of programme attendance did not differ between groups.

### 3.2. Interaction between Condition (Person- vs. Technique-Oriented) and Time (First and Third Academic Year) on Patient-Centeredness Attitudes, Interpersonal Behaviour, Empathy, and Perceived Communication Skills and Technical Knowledge

Regarding overall patient-centeredness attitudes (Figure 1), we found a significant main effect of time (F_(1,181)_ = 65.681, *p* ≤ 0.001, ⴄp^2^ = 0.266), condition (F_(1,181)_ = 27.445, *p* ≤ 0.001, ⴄp^2^ = 0.132), and time × condition interaction (F_(1,181)_ = 4.639, *p* = 0.033, ηp^2^ = 0.025) on the *PPOS Total* score scale. Regarding Caring and Sharing attitudes, we observed a similar main effect of time (*Caring*: F_(1,181)_ = 34.373, *p* ≤ 0.001, ⴄp^2^ = 0.160; *Sharing*: F_(1,181)_ = 58.725, *p* ≤ 0.001, ⴄp^2^ = 0.245) and condition (*Caring*: F_(1,181)_ = 17.959, *p* ≤ 0.001, ⴄp^2^ = 0.090; *Sharing*: F_(1,181)_ = 22.442, *p* ≤ 0.001, ⴄp^2^ = 0.110); however, we only identified a marginally significant effect of time × condition interaction effects on the *Sharing* variable (F_(1,181)_ = 3.815, *p* = 0.052, ⴄp^2^ = 0.021). Further pairwise comparisons revealed that overall patient-centeredness, Caring, and Sharing attitudes significantly improved in both groups from the 1st to the 3rd academic year (*Total scale*: person-oriented, *p* ≤ 0.001; technique-oriented, *p* = 0.001; *Caring*: person-oriented, *p* ≤ 0.001; technique-oriented, *p* = 0.019; and *Sharing*: person-oriented, *p* ≤ 0.001; technique-oriented, *p* = 0.002), and the three variables’ scores differed significantly between groups in the post-assessment measurement (*Total scale*: *p* = 0.001; *Caring*: *p* ≤ 0.001; and *Sharing*: ≤ 0.001).

We did not find significant time (F_(1,181)_ = 3.480, *p* = 0.064, ηp^2^ = 0.019), condition (F_(1,181)_ = 3.498, *p* = 0.063, ηp^2^ = 0.019), or time × condition interaction effects (F_(1,181)_ = 1.675, *p* = 0.197, ηp^2^ = 0.009) regarding empathy. However, the pairwise comparisons showed statistically significant differences between the first and the third academic years for the person-oriented group (*p* ≤ 0.001), suggesting a significant increase in empathy in students that attended person-oriented programmes across time; the same was not found in the technique-oriented group (*p* = 0.751). Additionally, we found that technique-oriented group scores for empathy were significantly lower than those for the person-oriented group at the third-year measurement (*p* = 0.016).

Regarding the Interpersonal behaviour scales (*Distress* and *Performance*), the analysis showed a significant main effect of time (*Distress*: F_(1,181)_ = 17.158, *p* ≤ 0.001, ηp^2^ = 0.087; *Performance*: F_(1,181)_ = 5.182, *p* = 0.024, ⴄp^2^ = 0.028) but not of condition (*Distress*: F_(1,181)_ = 0.348, *p* = 0.556, ηp^2^ = 0.002; *Performance*: F_(1,181)_ = 0.009, *p* = 0.924, ηp^2^ = 0.000). Also, we only found a significant main effect of time × condition interaction regarding the *Distress* scale (F_(1,181)_ = 5.795, *p* = 0.017, ηp^2^ = 0.031). In both the person-oriented and technique-oriented groups, the discomfort felt when they are required to act assertively increased significantly across time (person-oriented, *p* = 0.019; technique-oriented, *p* ≤ 0.001); however, students from technique-oriented programmes experienced a significantly greater increase in anxiety compared with students from the person-oriented group. Nevertheless, the mean scores did not significantly differ between groups in the third year (*p* = 0.552). No significant results were found in the pairwise comparison regarding the frequency in which groups of students act assertively, either between groups at the post-assessment measurement (*p* = 0.601) or for each group across time (person-oriented, *p* = 0.123; technique-oriented, *p* = 0.077).

Students’ mean perception of technical knowledge was significantly increased from the first to the third year for both the person-oriented (W = 274, Z = −8.228, *p* ≤ 0.001) and technique-oriented (w = 244, Z = −3.341, *p* ≤ 0.001) groups; at the post-assessment measurement, no significant differences were observed between groups (U = 2347, Z = −1.165, *p* = 0.244). On the other hand, the attendance of a person-oriented or technique-oriented programme did not significantly change students’ perception of their communication skills (person-oriented group: W = 1044, Z = −0.641, *p* = 0.521; technique-oriented group: W = 30, Z = −0.731, *p* = 0.465). However, we found a significant difference between groups regarding perception of communication skills in the post-assessment measurement, suggesting that students that attended person-oriented programmes had a significantly better perception of their communication skills than students from technique-oriented programmes in the third academic year (U = 2004, Z = −2.491, *p* = 0.013).

## 4. Discussion

The current study aimed to explore patient-centred attitudes, empathy, Interpersonal behaviours, and participants’ evaluation of their communication skills and technical knowledge in students attending healthcare programmes, which cover diagnostic, treatment, and rehabilitation areas. We assessed nursing, clinical physiology, physiotherapy, pharmacy, dietetics and nutrition, and medical imaging and radiotherapy students. We analyzed how these dimensions change between the first and third academic years, comparing the scores of students attending person- or technique-oriented professional programmes.

Considering the first assessment moment, performed at the beginning of the programme attendance, we found that overall health students’ patient-centred attitudes and empathy scores were moderate and high, respectively, considering the maximum score achievable. Similar findings were found at the commencement of the programme regarding both dimensions in studies that assessed nursing [19,43,60,62,63,64,65], physiotherapy [62], and pharmacy students [54,62]. Our sample is primarily female, and there is some evidence that female health students tend to have significantly higher scores in patient-centred attitudes [8] and empathy [20].

Also, the frequency of the assertive behaviour reported by students was superior to the Discomfort felt regarding the same behaviours for both students’ samples, suggesting that, overall, health students are *assertive*, according to Arrindell et al.’s [66] taxonomy. Similar results were found in a late Portuguese adolescent (16–21 years old) sample [46]. Although some studies show that female individuals report significantly higher distress when they behave assertively compared with males [46,67], the *Discomfort* mean scores achieved overall higher values in our sample (predominantly female, as previously noted). Further studies are needed to identify if there is any particularity in young people who apply for healthcare profession programmes that may increase anxiety in social contexts. Earlier contributions from cross-sectional studies also found moderate scores in communication apprehension in health students attending nursing [68,69], radiology [69], and pharmacy [68] programmes.

On the other hand, students from both professional programmes evaluated themselves as having moderate technical knowledge and good communication skills in the first academic year, considering the maximum score achievable. Using the same scale, Grilo et al. [60] found similar results in a sample with first-year nursing students. Although all the dimensions studied were assessed using self-report instruments, the students’ subjective perception of their communication skills appears to be congruent with the values observed concerning empathy, patient-centeredness, and assertiveness, as found in Kerr and Thompson’s [70] recent study with medical students.

It is also noticeable that, in the first academic year, students from person-oriented professional programmes had significantly higher scores in total and shared patient-centred attitudes than students who attended technique-oriented professional programmes. As such, these findings could suggest that health professions (such as nursing or physiotherapy) in which work depends more on establishing a close and trusting relationship with patients and their families (e.g., identifying the needs and preferences of the patient for better-adjusted recommendations and proposed treatments) might be more sought by individuals who have higher attitudes toward a patient’s centredness. Personality characteristics can affect an individual’s career choices and the type of work that most suits their values and attitudes [71]. Earlier studies found that health professionals from person-oriented professions scored higher than those with technique-oriented professions in several personality traits (e.g., *Cooperativeness* and *Self-Transcendence* in Campbell et al.’s study [36] and *Agreeableness* in Borges and Gibson’s work [72]). Cordina et al. [73] also evaluated the personality characteristics of new pharmacy students. The authors found that pharmacy students only scored higher in *personal relationships* traits, possibly showing a predisposition to provide care and develop collaborative relationships with pharmacy clients. The other traits (*Responsibility*, *Cautiousness*, and *Sociability*) scored lower, suggesting that the pharmacy profession might draw those attracted to following routine and conventional responsibilities, like drug distribution.

The stronger personality traits in individuals with person-oriented health professions are closer to more overall patient-centredness, sharing attitudes, and empathy. However, in our study, students who entered person-oriented and technique-oriented programmes in the first year did not significantly differ in their empathy scores. In a study with a sample of Portuguese medical students, it was found that *Openness to experience* and *Agreeableness* were the most relevant predictors of empathy when gender, age, and university were considered [74]. The admission requirements for the two different types of programmes considered in our study are the same and are not related to individual traits. Therefore, it is possible that this absence of differences in empathy but not in patient-centredness may be due to the individual characteristics of students and the possible existence of moderating factors that act differently in developing these two dimensions.

The descriptive analyses showed that, between the first and the third academic year, health students had an overall positive tendency to improve their patient-centred attitudes, empathy, and assertive performance, but also their discomfort when performing assertive behaviours during interactions. Patient-centred attitudes significantly increased from the first to the third year in both student groups; however, the PPOS *Total* and *Sharing* scores improved significantly more in students from person-oriented programmes. As mentioned, previous findings regarding changes in patient-centred attitudes throughout the programme have been reported mainly for specific health professions and through cross-sectional studies [75]. Although our results have suggested an increase in patient-centredness attitudes, it is valuable to extend our longitudinal research further on the academic programme, as the results of several longitudinal studies in medical schools pointed to a decrease in patient-centred attitudes after clinical training [9,11,12,13]. However, Tontus and Nebbiolo [76] observed a significant decrease in patient-centred attitudes from the first to the fourth year in a sample of Turkish medical students. One study [77] reported significant improvements in patient-centred attitudes, but only in the *Caring* dimension.

Concerning empathy, the pairwise comparison results suggest a significant increase from the first to the third year in the students’ person-oriented group; also, students who attended technique-oriented programmes scored significantly lower than students from person-oriented programmes in the third year. Nevertheless, no significant interaction time vs. condition effect was found for this dimension, suggesting that empathy evolution over time was not different in students of programmes with different orientations. It is a surprisingly positive result since most works carried out with nursing students and other allied health professions have reported a decline in empathy scores along the programme [19,78,79,80]. Similar conclusions were drawn in systematic reviews that included longitudinal and cross-sectional quantitative [16], qualitative studies [17], or both [18] with medical students [37]. The literature rarely compared empathy evolution in health programmes using person- vs. technique-oriented profession taxonomy. Blanco Canseco et al. [37] found significant differences in empathy levels between medical students who chose general and surgical-technological or non-clinical specialities. Several factors may be responsible for patient-centredness attitude improvement and empathy stability scores over time. However, the differences observed between groups might also be related to the relevant role of the academic curriculum, which is more or less focused on the technical aspects of healthcare depending on the orientation of the health profession.

Regarding students’ reported performance of assertive behaviours, although we found an overall time effect, there were no significant differences in the changing pattern between students from the person- and technique-oriented programmes across time, nor did the scores achieved in the third academic year differ significantly between groups. Our findings did not corroborate previous studies with Greek and Irish nursing students that identified increased assertiveness across programme attendance [30,31]. The results also suggest that the discomfort experienced when performing assertive behaviours increased in both groups, being significantly more accentuated in students from technique-oriented programmes. At university, students will be exposed to challenging situations where they must assert their identity and needs while seeking social integration [81]. Therefore, the general increase in assertiveness-related anxiety might be related to the adaptive tasks of late adolescence and entry into adulthood. On the other hand, students who enrol in person-oriented programmes may have an increased ability (or more effective skills) to emotionally manage situations in which they have to act assertively due to their personality characteristics [73] or even programme-specific interactions and learning experiences that are naturally more oriented to the particularities of interactions with patients, communication, and the therapeutic relationship. LaRochelle and Karpinski [82] also found moderate to high scores in communication apprehension in several fourth-year pharmacy students’ samples. These results deserve particular attention and should continue to be studied in the last year of the programme since experiencing very high anxiety rates during communication, in general, can negatively impact how students communicate with patients and their families. Although students have some experiences in hospital and primary care contexts in the programmes selected for this study, many curricular internships take place in the last academic year, where interaction with patients is quite intensive.

The students’ perceived communication skills did not significantly improve across the programme attendance for any of the groups; however, we found a significant difference between groups in the third year, with students from person-oriented programmes reporting a significantly higher perception of communication skills than students from technique-oriented programmes. On the contrary, the perception of technical knowledge significantly improved for both student groups. As expected, this pattern of results was also found in cross-sectional studies with medical [61] and nursing [60] students. However, only the study with medical students showed no significant differences across time regarding students’ perceived communication skills [61]. We can hypothesize that developing patient-centred communication skills is more highly valued by students from person-oriented programmes who have more significant positive attitudes about it. As a result, they might invest more time in enhancing their communication skills. On the other hand, in person-centred programmes, curriculum communication skills are more intensively taught. Therefore, based on Bandura’s social learning theory [83], it is conceivable that students increased their confidence in their communication skills and reinforced their attitudes about patient-centredness. Finally, the differences regarding students’ perceptions of technical knowledge and communication skills over their academic years could indicate that the achievement of these skills is perceived in distinct ways [61].

## 5. Limitations

Our results should be read considering several limitations. Although this is a longitudinal study with nursing and allied health students, which is an added value for the research in this field, our sample is based on convenience sampling with two Portuguese public health schools. Also, the self-selecting nature of the sample demands prudence since, although the study was presented to all students that year, it is possible that those who agreed to take part are also those who are already most prone to person or technique-oriented areas. Additionally, the study only contemplated nursing and allied students who started their studies at the beginning of the 2016/17 academic year; that is, only one group of students for each programme was contemplated. Although all assumptions were confirmed before carrying out inferential tests, the results of the person- and technique-oriented profession comparison, both regarding the baseline and across time, must be interpreted with caution, as the imbalance in sample size reduced the power of the tests. Likewise, the students who agreed to participate in the study by answering the questionnaires in the first and third years were mainly women (88%). Although most undergraduate students in healthcare programmes in Portugal are women, our sample was not stratified, which prevents the generalization of the results. Finally, we used self-report surveys, which may compromise the reliability of the students’ responses due to the social desirability predisposition. Future studies that include more nursing and health schools and extending longitudinal studies to include clinical years and observational assessment measures besides self-report surveys are recommended.

## 6. Conclusions

Our findings suggest that, overall, Portuguese students who applied for nursing and allied academic programmes had moderate to high patient-centredness attitudes, empathy and assertiveness scores, and perceived themselves as having good communication skills; however, they experienced discomfort associated with assertive behaviours from the first to the third academic year, with a more accentuated increase in students of technique-oriented programmes. The results suggest that these nursing and allied programmes attract young people with an adequate profile to carry out a learning process on healthcare activities, but also highlight the necessity of being more attentive to students’ emotional states when interacting with patients, families, and colleagues in a multidisciplinary team.

## Figures and Tables

**Figure 1 healthcare-11-02803-f001:**
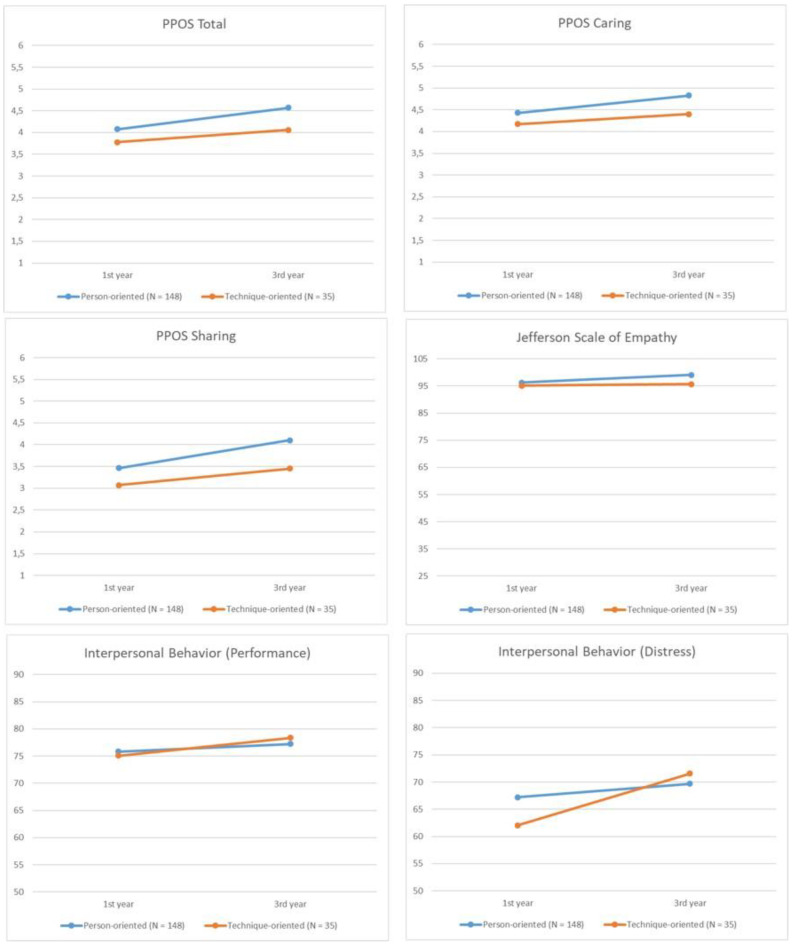
*Mean scores for* patient-centeredness attitudes (Total, Sharing, and Caring), Empathy, and Interpersonal behaviour (Performance and Distress) in the *1st and 3rd academic years for students attending person-oriented and technique-oriented profession* programmes.

**Table 1 healthcare-11-02803-t001:** Sociodemographic characteristics of the students (whole sample, person-oriented, and technique-oriented).

Sociodemographic Characteristics	Whole Sample		Person-Oriented Group		Technique-Oriented Group	
	*n*	%	*n*	%	*n*	%
Sex ^a^						
Male	11	6.0	11	7.4	0	0.0
Female	161	88.0	130	87.8	31	88.6
Marital status						
Single	181	98.9	146	98.6	35	100
Married	2	1.1	2	1.4	0	0.0
Working student ^b^						
Yes	12	6.6	11	7.4	1	2.9
No	167	91.3	135	91.2	32	91.4
Displaced from their hometown to study						
Yes	60	32.8	48	32.4	12	34.3
No	123	67.2	100	67.6	23	65.7
Programmes						
Clinical physiology	14	7.7			14	100
Dietetics	13	7.1	13	100		
Imaging and Radiotherapy	15	8.2			15	100
Nursing	109	59.6	109	100		
Pharmacy	6	3.3			6	100
Physiotherapy	26	14.2	26	100		

Legend: ^a^ Missing values regarding *Sex* for 11 (whole sample), 7 (person-oriented group), and 4 (technique-oriented group) participants. ^b^ Missing values regarding *Working student* for 4 (whole sample), 2 (person-oriented group), and 2 (technique-oriented group) participants.

**Table 2 healthcare-11-02803-t002:** *Descriptive analysis of the variables* (patient-centeredness attitudes, Interpersonal behaviour, Empathy, and Perceived communication and technical skills) *assessed in the 1st and 3rd academic year (whole sample, person-oriented, and technique-oriented groups)*.

Measure	Whole Sample (N = 183)				Person-Oriented (N = 148)				Technique-Oriented (N = 35)			
	1st Year		3rd Year		1st Year		3rd Year		1st Year		3rd Year	
	M	SD	M	SD	M	SD	M	SD	M	SD	M	SD
PPOS Total	4.02	0.50	4.47	0.52	4.08	0.49	4.57	0.46	3.78	0.44	4.06	0.57
PPOS Caring	4.38	0.53	4.75	0.54	4.43	0.52	4.83	0.47	4.17	0.55	4.40	0.65
PPOS Sharing	3.39	0.71	3.97	0.70	3.46	0.70	4.10	0.65	3.07	0.68	3.45	0.70
Jefferson Scale of Empathy	96.04	8.55	98.44	7.59	96.25	8.71	99.10	7.21	95.14	7.92	95.66	8.61
Interpersonal Behaviour (Distress)	66.21	16.76	70.07	16.60	67.19	17.16	69.71	16.97	62.06	14.45	71.57	15.04
Interpersonal Behaviour (Performance)	75.68	11.50	77.45	11.50	75.83	11.82	77.24	11.69	75.06	10.14	78.37	10.77
	Me	IQR	Me	IQR	Me	IQR	Me	IQR	Me	IQR	Me	IQR
Perceived Communication Skills	4	1	4	0	4	0	4	0	4	1	4	1
Perceived Technical Knowledge	3	2	4	0	3	2	4	0	3	2	4	1

Legend: M (mean), SD (standard deviation), Me (median), IQR (interquartile range).

## Data Availability

Data used and analyzed during the current study are available from the corresponding author upon reasonable request.

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
