# Peer review of "Attitudes toward Patient-Centred Care, Empathy, and Assertiveness among Students in Rehabilitation Areas: A Longitudinal Study"

_healthcare, 2023, doi:10.3390/healthcare11202803_

Round 1

Reviewer 1 Report

As a professor in a medical school, intuitively, I would agree with the results. Medical students (and in close specialties, such as nurcing) make notable progress in their attitude to patients (patient-centred care, empathy, and assertiveness), especially from the 1st to the 3rd academic year.
It would be interesting to know from further studies by this group how this progress proceeds (or do not proceed) within following years (4-6th academic year) and even further.
Discussion provides comparison with students of other regions (states) which supports the outcome.
In a whole, I do not have major comments.
Maybe, figure could be improved (colour looks too pale, and inscriptions in the figure are too small). Why not to show SD values of mean?

Author Response

Dear Reviewer
Herewith we enclosed the revised manuscript to be submitted to the Healthcare, on the Special Issue Clinical Communication in Rehabilitation. entitled "Attitudes toward patient-centred care, empathy, and assertiveness among students in rehabilitation areas: a longitudinal study".
We are grateful for the helpful comments and questions raised. We have attended to these issues above, hoping our answers adequately address your comments. All the modifications in the manuscript are highlighted in red colour.
We thank the reviewers' contribution, which helped improve the manuscript’s quality. We hope this version complies with the Healthcare standards for publication.

Reviewer 2 Report

Manuscript Healthcare

Title: Attitudes toward patient-centred care, empathy, and assertiveness among students in rehabilitation areas: a longitudinal study

Thank you to the authors for an interesting and well referenced study report. The manuscript is verbose, as written in thesis style, and requires significant rework to be publishable.

Firstly, the Introduction wordy and repetitive and hides great concepts, as well as the study justification in verbosity. The first three paragraphs restate the aspects of patient centred care from the points of view of generic, physical therapy and rehabilitation medicine. While this reinforces the importance, it is written in a thesis rather than manuscript style, and it not necessary to labour the point in this way. Please markedly rationalize and shorten the Introduction by reducing the quotations and justifying the study only once. This reviewer strongly suggests that the Introduction start at paragraph 4, omitting the text prior and references from the first three included to bolster the scientific support for the statements there. Similarly, paragraphs 5 – 8 present the concept of empathy but with significant reinforcement and duplication. Also, assertiveness is laboured in paragraphs 9 and 10 to be reinforced by a subsection discussing both empathy and assertiveness in students as if it is a different concept in that context. Please consider omitting paragraphs 5 – 10 and references added to the subsection discussing students. The journal is titled ‘Healthcare’ – concepts such as these are not foreign and do not need to be laboured, by nature of the types of publications accepted and readership targeted. Please shorten Introduction by minimum half the word count.

Secondly, Aims should be stated clearly, in one sentence and without further justification – as all of which occurs, or should be found in Introduction. Any scientific arguments and supporting references in the Aims section should be rationalized into Introduction.

In Materials and Methods – significantly debulk the text throughout, and improve further by:

1.       Moving information about the participant sample to Results.

2.       Please move Procedure to precede Instruments section.

3.       In Statistical Analysis, explain and justify how missing data and exclusions were dealt with. Also were there any adjustments made for the unbalanced group numbers (low numbers in technique-orientated group)?

In Results, debulk the section by not restating / duplicating in text results presented in tables.

Please consider in Discussion that enrolment processes and requirements are likely to have biased significantly, the baseline characteristics of the participants. Therefore, arguments in relation to baseline changes, must be tempered to reflect that fact. Also, the analysis is underpowered to make statements about equivalence and all statements comparing results between technique and person oriented course groups.

The Conclusion must be markedly debulked to only include primary take home messages of the study.

Well referenced.

Finally, please address and respond to each specific point (n=30) marked up in comments in the PDF file.

English is good with minor grammatical adjustments required.

Author Response

Dear Reviewer
Herewith we enclosed the revised manuscript to be submitted to the Healthcare, on the Special Issue Clinical Communication in Rehabilitation. entitled "Attitudes toward patient-centred care, empathy, and assertiveness among students in rehabilitation areas: a longitudinal study".
We are grateful for the helpful comments and questions raised by the reviewers. We have attended to these issues above, hoping our answers adequately address the reviewers’ comments. The comments of the reviewer made in the text and the PDF were combined in the table for a formal response. All the modifications in the manuscript are highlighted in red colour.
We thank the reviewers' contribution, which helped improve the manuscript’s quality. We hope this version complies with the Healthcare standards for publication.
Sincerely yours,